# A dual computational and experimental strategy to enhance TSLP antibody affinity for improved asthma treatment

**Yitong Lv[1], He Gong[1], Xuechao Liu[2], Jia Hao[2], Lei Xu[2], Zhiwei Sun[2], Changyuan Yu[1]*,
Lida Xu[2,3]***

**1** College of Life Science and Technology, Beijing University of Chemical Technology, Beijing, China,
**2** Beijing Sungen Biomedical Technology Co., Ltd, Beijing, China, **3** Beijing Hotgen Biotech Co., Ltd, Beijing,
China

\* yucy@mail.buct.edu.cn (CY); lida.xu@hotgen.com.cn (LX)

Singapore, SINGAPORE

**Data Availability Statement:** All relevant data are
within the manuscript and its Supporting
Information files.

## Abstract

Thymic stromal lymphopoietin is a key cytokine involved in the pathogenesis of asthma and
other allergic diseases. Targeting TSLP and its signaling pathways is increasingly recog-
nized as an effective strategy for asthma treatment. This study focused on enhancing the
affinity of the T6 antibody, which specifically targets TSLP, by integrating computational and
experimental methods. The initial affinity of the T6 antibody for TSLP was lower than the
benchmark antibody AMG157. To improve this, we utilized alanine scanning, molecular
docking, and computational tools including mCSM-PPI2 and GEO-PPI to identify critical
amino acid residues for site-directed mutagenesis. Subsequent mutations and experimental
validations resulted in an antibody with significantly enhanced blocking capacity against
TSLP. Our findings demonstrate the potential of computer-assisted techniques in expediting
antibody affinity maturation, thereby reducing both the time and cost of experiments. The
integration of computational methods with experimental approaches holds great promise for
the development of targeted therapeutic antibodies for TSLP-related diseases.

## Author summary

Computer-assisted affinity maturation significantly reduces experimental time and lowers
research costs. Targeting thymic stromal lymphopoietin and its signaling pathways with
specific antibody drugs is widely recognized as an effective strategy for treating asthma. In
our study, we successfully identified a TSLP-targeting antibody from the fully synthetic
human phage antibody libraries. We integrated computer-assisted methods to enhance
the antibody's affinity. These techniques enabled efficient prediction of critical amino acid
residues, guiding targeted mutagenesis experiments. By combining computer-assisted
approaches with experimental methods, we have successfully developed a mature method
for enhancing antibody affinity. Through this research, we have obtained an antibody
with high affinity for TSLP, providing a new avenue for treating asthma and other TSLP-

**Funding:** This work was supported by National Natural Science Foundation of China (82174531 to CY), National High Level Hospital Clinical Research Funding (XK2023-13 to YL) and Scientifc and Technological Research Project of Xinjiang Production and Construction Corps (2022AB022 to HG). The funders had no role in study design, data collection and analysis, decision to publish, or preparation of the manuscript.

**Competing interests:** I have read the journal's policy and the authors of this manuscript have the following competing interests: Authors Xuechao Liu, Jia Hao, Lei Xu, Zhiwei Sun and Lida Xu are employed by the company Beijing Sungen Biomedical Technology Co., Ltd. Lida Xu is also employed by Beijing Hotgen Biotech Co., Ltd. The remaining authors declare that the research was conducted in the absence of any commercial or financial relationships that could be construed as a potential conflict of interest.

related diseases. The computer-assisted affinity maturation strategy brings hope for speeding up the drug development process.

## Introduction

The cytokine thymic stromal lymphopoietin (TSLP), derived from epithelial cells, is involved in the initiation and persistence of asthma inflammatory pathways [1,2]. It has been found that TSLP forms a trimeric signaling complex with the thymic stromal lymphopoietin receptor (TSLPR) and Interleukin-7 receptor alpha chain (IL-7Rα), activating intracellular signaling via the STAT5 pathway [3–5], which leads to the release of inflammatory cytokines. Targeted antibody drugs against TSLP and its signaling pathway are considered effective strategies for asthma treatment [6].

Tezepelumab (AMG 157) is a fully human monoclonal antibody (immunoglobulin G2λ) that specifically targets TSLP, hindering its interaction with the TSLP receptor complex and effectively inhibiting multiple downstream inflammatory pathways [7]. X-ray crystallography studies have identified the epitope binding sites between AMG157 and TSLP, revealing that AMG157 occupies the binding interfaces of TSLP and TSLPR, disrupting their interaction [3]. Based on the success of AMG157, our research aims to identify a novel antibody that not only binds effectively to TSLP but also possesses a higher affinity than AMG157, thereby blocking the activation of downstream pathways by TSLP.

Antibodies undergo affinity optimization before they can be considered potential therapeutic drugs [8]. Techniques such as site-directed mutagenesis, chain shuffling, and error-prone PCR are commonly utilized for antibody affinity maturation [9,10]. However, this process is often time-consuming, spanning several months. The advancement in computational power has facilitated the development of various strategies for guiding the rational engineering of antibody binding and specificity. In silico approaches such as structure-based and mini-library methods have played a crucial role in antibody affinity maturation by enabling the exploration and optimization of antibody-antigen interactions [8,11,12]. These techniques rely on high-quality co-crystal structure and algorithms capable of accurately computing the energy variations resulting from mutations. The development of machine learning and deep learning has opened new avenues for affinity maturation. Tools like mCSM-PPI2 and Geo-PPI integrate multiple factors, including graph-based signatures and atomic interactions, to predict the effects of mutations on the antibody-antigen affinity [13–15]. These tools have proven valuable in analyzing single-point and multi-point mutations and providing insights into changes in affinity. By utilizing these software tools, we aim to establish a computer-assisted approach for accelerating the maturation of antibody affinity.

In this study, we applied a methodology that combines computational and experimental approaches to accelerate the process of affinity maturation for antibody targeting TSLP. Initially, we screened our fully synthetic human phage antibody libraries and successfully identified T6, a specific antibody that targets TSLP. Subsequently, we employed experimental alanine scanning to identify critical amino acids, ensuring the accuracy of the T6-TSLP complex structure. Additionally, we utilized tools such as mCSM-PPI2 for site-directed mutagenesis on key amino acids and validated the affinity enhancement strategy through GEO-PPI. Ultimately, we obtained an antibody that demonstrated superior binding affinity to TSLP compared to AMG157, as confirmed through cell blocking assays.

## Results

### Model construction of TSLP-T6 complex

We obtained an antibody named T6 that targets TSLP using fully synthetic human antibody libraries. To determine the crucial amino acid residues of the T6 antibody, we performed alanine scanning experiments, and the results are summarized in Table 1. Additionally, we conducted enzyme-linked immunosorbent assay (ELISA) experiments to assess whether the binding epitopes between T6 and AMG157 on TSLP are consistent. AMG157 was coated on the plate, and 10 ng/mL of TSLP and different concentrations of antibodies were added to detect the signal of the AMG157-TSLP complex using an ELISA reader. As shown in Fig 1A, as the concentration of free AMG157 antibody increased, the amount of TSLP bound to the coated AMG157 gradually decreased, resulting in a gradual decrease in signal values. Under the same antibody concentration, upon adding T6 antibody after the saturation of AMG157 and free TSLP binding, a significant decrease in signal value was observed. This observation suggests notable differences in the binding epitopes of T6 and AMG157 antibodies on TSLP.

The complex structure of TSLP and T6, as depicted in Fig 1B, was obtained using the ZDOCK docking method, and their binding sites were analyzed with Ligplot+ [16]. Fig 1C demonstrated that the key amino acids (27L, 29D, 31Y, 49E, 97L in the light chain; 52S, 57S, 59Y, 99D, 102W in the heavy chain) determined through alanine scanning experiments were located at the binding interface of the model, confirming the accuracy of the docking model. This model served as the basis for subsequent site-directed mutagenesis. Moreover, we analyzed the binding sites of TSLP with its receptors TSLPR and IL-7Rα (PDB ID: 5J11), as well as the binding sites of TSLP with AMG157 (PDB ID: 5J13). The results revealed that AMG157 occupied the binding sites of TSLP with TSLPR, while T6 occupied a distinct position on TSLP (Fig 1C). Specific details about the binding sites are provided in S1 Fig and S1 Table.

### Cellular functional assessment of T6 antibody

Upon binding to its receptors, TSLP activates signaling pathways, including the phosphorylation of STAT5, which serves as a key marker [17]. To evaluate pathway activation, we transfected CHO cells with TSLPR, IL-7Rα, and STAT5, and observed a concentration-dependent increase in phosphorylated STAT5 levels upon TSLP stimulation, confirming pathway activation. Subsequently, we added T6 antibody, and as the concentration of T6 increased, the detection signal gradually decreased, indicating its ability to block the pathway (Fig 2).

**Table 1. Key amino acid residues in alanine scanning of the antibody.**

| Position | Mutations |
|---|---|
| L-CDR1 | T6L-L27A |
| L-CDR1 | T6L-K30A |
| L-CDR1 | T6L-Y31A |
| L-CDR1 | T6L-A32G |
| L-CDR2 | T6L-D50A |
| L-CDR3 | T6L-W90A |
| L-CDR3 | T6L-L97A |
| H-CDR2 | T6H-S52A |
| H-CDR2 | T6H-S57A |
| H-CDR2 | T6H-Y59A |
| H-CDR3 | T6H-D99A |
| H-CDR3 | T6H-W102A |
| H-CDR3 | T6H-F105A |

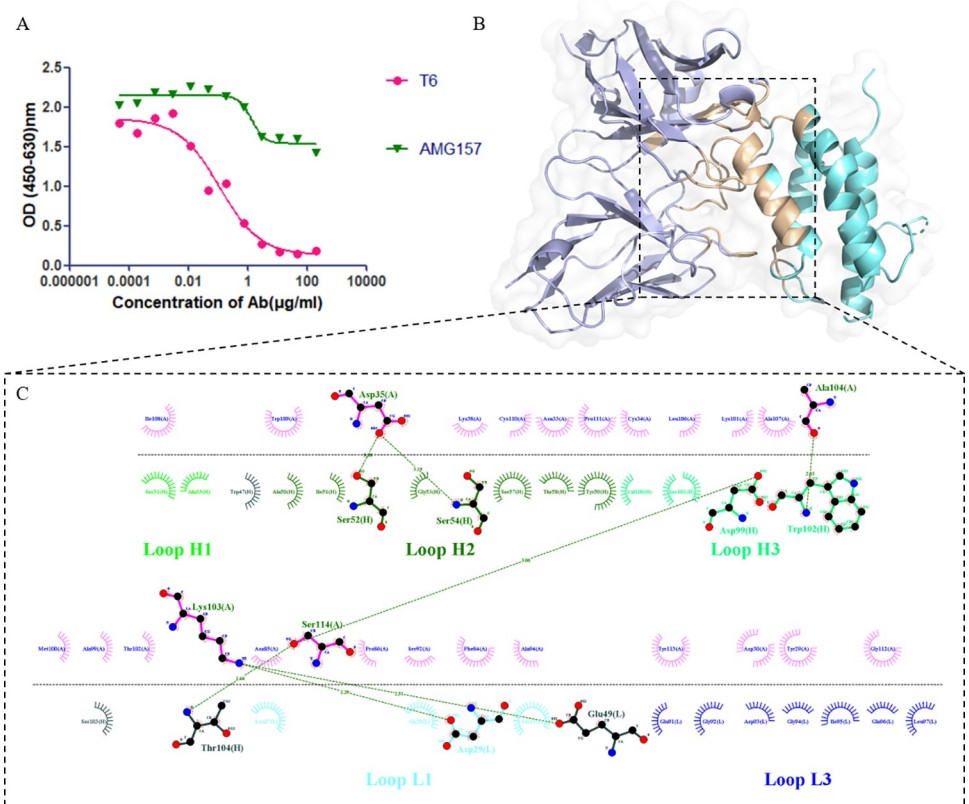

**Fig 1. Structure and interaction of the TSLP-T6 complex.** (A) ELISA experiments determining the binding epitopes of AMG157 and T6 on TSLP. (B) Complex structure of TSLP (shown in aquamarine) and T6 (shown in light blue). The binding interface is depicted in wheat color. (C) Ligplot+ analysis of the interactions between TSLP and T6.

Furthermore, the AMG157 antibody displayed a slightly superior blocking efficacy compared to that of T6. Consequently, we optimized the T6 antibody's affinity for TSLP, thereby enhancing its blocking capability.

### Single-point mutations of T6 antibody

We utilized mCSM-PPI2 and FoldX tools to perform alanine scanning on our model, and the results were presented in S2 and S3 Tables. Residues that exhibited absolute binding free energy changes greater than 1 (predicted by mCSM-PPI2) and absolute changes exceeding 5 (predicted by FoldX) were considered significant. By intersecting the identified residues with experimental alanine scanning results (Table 2), the critical amino acids were identified.

For the identified key amino acids, we conducted single-point mutation predictions. Each residue was mutated to the remaining 19 amino acids, and the changes in binding free energy after mutation were predicted using mCSM-PPI2. We focused on the sites predicted to enhance binding affinity, as indicated in S4 Table. Subsequently, the generated mutation schemes were screened using the criteria outlined in Section In silico mutagenesis. The selected mutation schemes were further validated using GEO-PPI. Ultimately, we chose 16 mutation schemes, and functional assays were performed to evaluate the blocking efficacy of these 16 mutated antibodies, as depicted in Fig 3.

In general, the blocking efficacy of an antibody refers to its ability to prevent the interaction between a target molecule and its receptor, thereby inhibiting downstream signaling pathways.

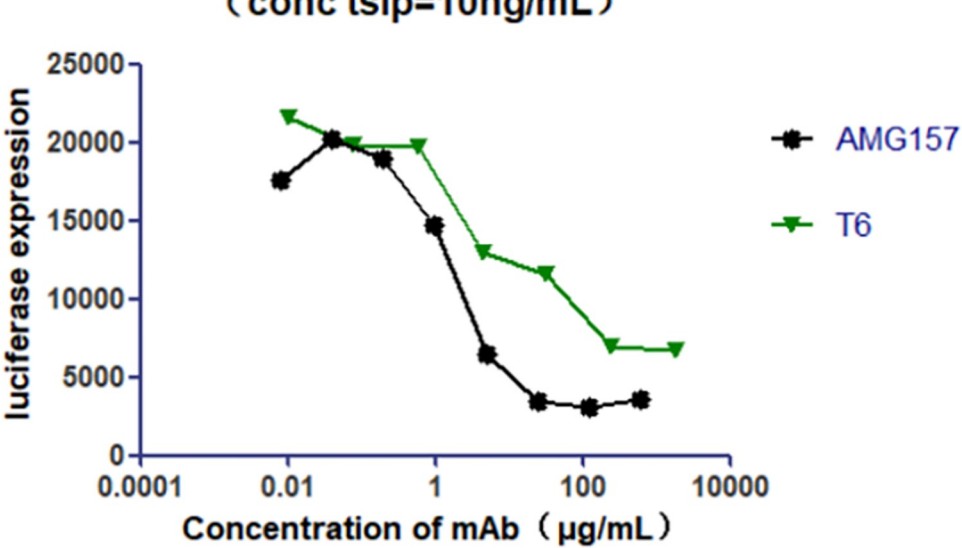

**Fig 2. Validation of cell-blocking function of T6 antibody.**

The strength of binding between an antibody and its target, such as TSLP, is referred to as affinity. The better the blocking efficacy of an antibody, the higher its affinity with the target molecule. Based on the data presented in Fig 3, it is evident that the mutation of light chain residue 49GLU to TYR (L49Y) significantly enhanced the blocking efficacy of the mutant strain and approached the level of AMG157. This suggested that the L49Y mutant has a higher affinity for TSLP compared to T6.

IC50 (half maximal inhibitory concentration) refers to the concentration of the measured antagonist that produces half of the maximum inhibition. Although the L49Y mutant showed an improvement in blocking efficacy, with an IC50 value of 0.6921 (S5 Table), while AMG157 had an IC50 value of 0.5771, it is important to note that the IC50 value of the L49Y mutant strain remained slightly lower than that of AMG157. Similarly, mutations L49F, L90Y, and L97F also demonstrated improved affinity compared to T6. Notably, these affinity-enhancing antibodies primarily involved mutations to hydrophobic amino acids, highlighting the importance of hydrophobic interactions in mediating the binding affinity between the antibody and the antigen. Based on the promising results of the L49Y mutation, a subsequent second round of mutations was performed, building upon the L49Y variant.

## Analysis of interaction strength and energy changes in T6-L49Y

We introduced a mutation in the T6 antibody by replacing the glutamic acid at position 49 of the light chain with tyrosine (L49Y). To ensure the stability of the T6-L49Y mutant structure, a 100 nanosecond molecular dynamics (MD) simulation was performed, and a stable stage of

**Table 2. Key amino acids for initial round of mutations.**

| Chain | Residues |
|---|---|
| H | 52,53,57,58,59,99,102,105 |
| L | 27,30,31,32,33,49,50,90,91,92,93,97 |

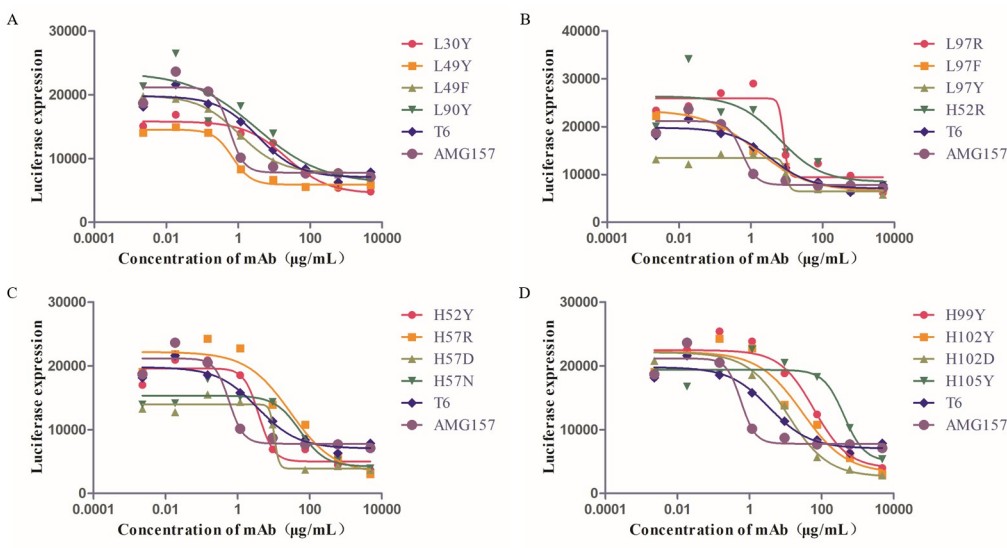

**Fig 3. Antibody blocking efficacy after the first round of single-point mutations.**

the system was achieved at approximately 70 nanoseconds during the simulation, as shown in Fig 4A.

To investigate the dynamic behavior, we extracted the MD trajectory segment spanning from 70 to 100 nanoseconds and conducted root-mean-square fluctuation (RMSF) analysis. Notably, significant fluctuations were observed in the RMSF values at positions 50–70 of the heavy chain, and 20–30 and 90–108 of the light chain, indicating highly flexible regions. To gain further insights into the structural changes, we aligned the stable conformations extracted from this trajectory segment with the initial conformation. Our analysis revealed minimal alterations, with only minor fluctuations in the secondary structure of these three segments, as

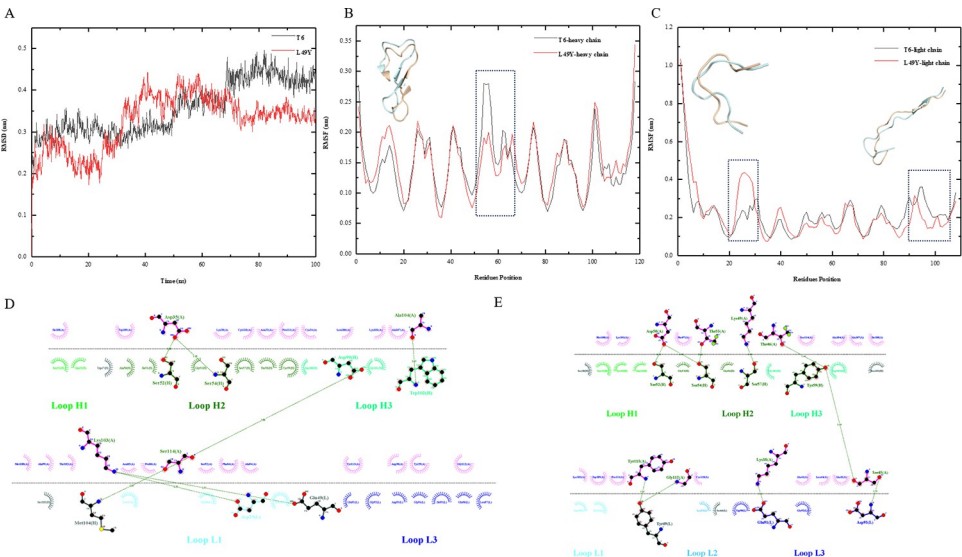

**Fig 4. Dynamic simulation and force analysis graphs.** (A) RMSD plot of the L49Y dynamic simulation; (B) RMSF plot of the L49Y heavy chain dynamic simulation; (C) RMSF plot of the L49Y light chain dynamic simulation; (D) Force analysis plot of the T6-TSLP interaction; (E) Force analysis plot of the L49Y-TSLP interaction.

**Table 3. Key amino acids for the second round of mutations.**

| Chain | Residues |
|---|---|
| H | 33,52,57,59,62,99,102,103 |
| L | 29,50,90,91,92,93,95,97 |

depicted in Fig 4B and 4C. These findings suggested that despite the dynamic nature, the overall secondary structure of the T6-L49Y mutant remained relatively stable throughout the simulation.

Next, we investigated the effects of the L49Y mutation on the energy landscape of the T6 antibody. Using the FoldX software, we analyzed the energy changes before and after the mutation, and the results were summarized in S6 Table. Overall, the T6 antibody exhibited minimal energy changes upon the L49Y mutation, with slightly lower energy values observed after the mutation compared to the wild type. Additionally, we analyzed the changes in the interactions at position 49 of the light chain before and after the mutation (Fig 4D and 4E). The mutation of position 49 to tyrosine resulted in a shift in the interactions with the antigen from hydrogen bonding to hydrophobic interactions, which may have contributed to maintaining the stability of antigen-antibody complexes in biological systems.

## Second round docking and site-directed mutagenesis

In the first round of mutations, the L49Y mutation was identified as enhancing blocking efficacy. Therefore, in the second round of mutations, single-point mutations were performed based on the L49Y mutant strain. Table 3 presented the key amino acids in the T6-L49Y-TSLP complex structure. Single-point mutations were performed on these key amino acids. Following the screening criteria outlined in Section In silico mutagenesis, mutations predicted to increase affinity by mCSM-PPI2 were selected (S7 Table). These selected mutations were further validated using GEO-PPI. Fig 5 illustrated the result of functional assays conducted on the 19 selected mutated strains. It was observed that the mutation of leucine to aspartate at

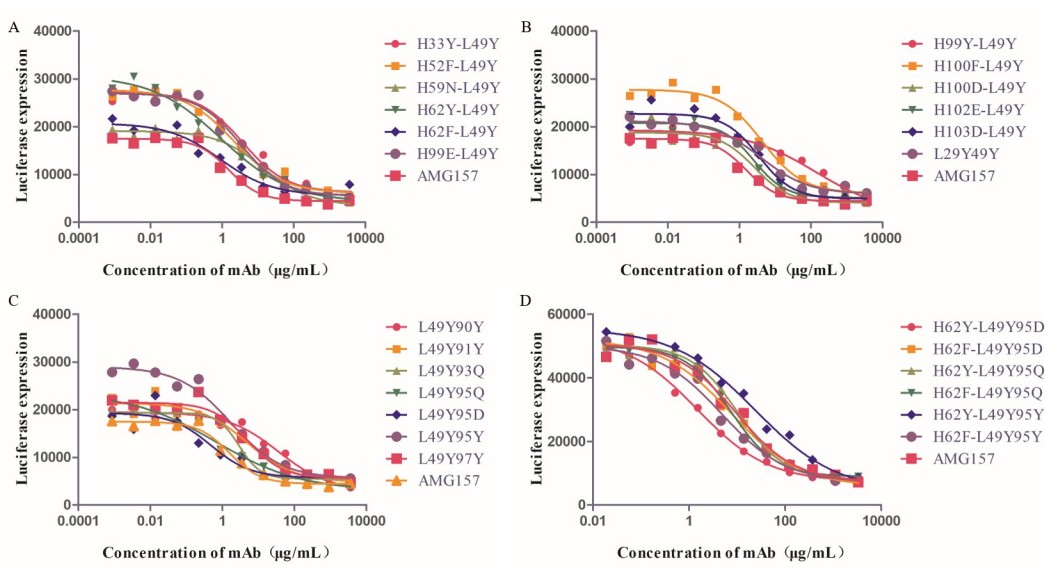

**Fig 5. Antibody blocking efficacy after the second round of single-point mutations.**

position 95 (L95D) in the light chain exhibited a higher blocking efficacy than AMG157, with an IC50 value of 0.4777 compared to AMG157's IC50 value of 1.54. Comparing the IC50 values of the various mutated strains, we observed that the H62Y, H62F, and L95Q mutations had IC50 values of 0.9479, 0.8978, and 0.6455, respectively, demonstrating slightly higher blocking efficacy compared to AMG157. These findings were consistent with the data presented in S8 Table. Additionally, the L95Y mutant strain exhibited an IC50 value of 1.696, which was similar to the IC50 value of AMG157 (1.54). Therefore, the L95Y mutation was also considered as a potential candidate for our future combination strategies.

Consequently, we performed pairwise combinations of the favorable mutation schemes obtained in the second round of mutations based on the L49Y mutant strain. Subsequently, we expressed the mutated strains resulting from these pairwise combinations and assessed their blocking efficacy. Through this process, we discovered that the H62Y-L49Y95D mutated strain exhibited the optimal blocking efficacy (S9 Table). Additionally, the H62F-L49Y95D, H62F-L49Y95Q, and H62F-L49Y95Y mutated strains showed slightly improved blocking efficacy compared to AMG157. Overall, based on the functional assay results presented in Fig 5D, the H62Y-L49Y95D mutated strain was confirmed as the most effective in terms of blocking efficacy.

## Discussion

Targeting TSLP and its associated signaling pathways has emerged as an attractive strategy for the treatment of asthma. The objective of this study is to computationally enhance the affinity of antibodies targeting TSLP. We employed a combination of docking modeling, alanine scanning experiments, and software tools for mutagenesis predictions. Through these methods, we successfully obtained a high-affinity antibody against TSLP and established a cost-effective process for antibody affinity maturation. Our findings highlight the significance of hydrophobic interactions and emphasize the importance of considering the stability of the antibody-antigen complex during the mutagenesis process. By integrating computational methods, we expedited the antibody optimization process and demonstrated the potential of computer-assisted techniques in enhancing antibody affinity.

The construction of the model is a crucial first step in affinity maturation research, and therefore, the accuracy of the model is of utmost importance. The framework regions of antibodies are highly conserved, making them easy to model, while the CDR (complementarity-determining region) loops are highly variable and require additional constraints for accurate modeling [18]. Modeller and SWISS-MODEL remain the most commonly used software for antibody modeling [19,20]. Modeller is a user-friendly homology modeling software that is highly effective when suitable template structures with high similarity and coverage can be retrieved from protein databases [21]. With the continuous iteration and improvement of algorithms, the accuracy of modern modeling software has also been steadily increasing. In recent years, modeling methods such as Rosetta Antibody [22], RoseTTAFold [23], AlphaFold [24], and DeepAb [25] have also made significant improvements in accuracy, providing robust support for computer-assisted affinity maturation research.

In our comprehensive approach, we utilized two tools, mCSM-PPI2 and Geo-PPI, to predict and optimize the performance of antibodies within computer-aided affinity maturation platforms. Initially, we conducted alanine scanning experiments to identify key amino acids, ensuring their presence at the interface of the computational docking model and validating the accuracy of the antigen-antibody docking model. To predict the impact of antibody mutations on affinity, we employed mCSM-PPI2, a software that utilizes graph-based structural features [13]. This method enables the simulation of spatial interactions, as well as geometric and

physicochemical properties of the complexes [26,27]. However, protein-protein interactions are highly complex, and besides the known features, there may exist other correlations that are not directly observable. To address this limitation, we incorporated Geo-PPI, which employs a geometric encoder to automatically learn meaningful features from protein structures [15]. Geo-PPI employs a supervised training gradient boosting tree (GBT) approach to learn the mapping between mutation geometry representations generated by the geometric encoder and the corresponding mutation effects. This enhances its generalization capability when dealing with unseen complexes. It should be noted that Geo-PPI can only predict a single amino acid mutation at a time, while mCSM-PPI2 allows for the simultaneous mutation of an amino acid at a given position to any of the other 19 amino acids. Therefore, in our approach, we initially used mCSM-PPI2 to predict the mutagenesis sites, resulting in multiple potential mutation options. Subsequently, we combined Geo-PPI to further filter and predict the most promising mutation schemes, which were subsequently experimentally validated. By combining the strengths of mCSM-PPI2 and Geo-PPI, we were able to leverage the high-throughput mutation prediction capability of mCSM-PPI2 and utilize Geo-PPI to provide more accurate and generalized prediction results.

Our findings from the mutagenesis process revealed several important factors that contribute to enhancing antibody affinity. Firstly, we observed a significant increase in antibody affinity by replacing the aspartate residue at position 49 on the light chain with tyrosine. This substitution resulted in a shift from hydrogen bond interactions to hydrophobic interactions, highlighting the importance of hydrophobic interactions in stabilizing the antibody-antigen binding. This finding aligns with previous studies that have demonstrated the role of hydrophobic interactions in antibody-antigen recognition and binding stability. Additionally, during the mutagenesis process, we considered the distance between amino acids at the binding interface as an important principle to ensure successful mutagenesis. By applying this principle, we conducted a second round of mutagenesis and identified several mutations, including H62Y, H62F, L95Y, L95D, and L95Q. Although these mutations only resulted in modest improvements in affinity, the inclusion of hydrophobic amino acids, such as tyrosine, highlighted the significance of hydrophobic interactions in maintaining stable antigen-antibody binding.

In the end, we successfully obtained an antibody that exhibits high-affinity binding to TSLP. Compared to the control antibody AMG157, this antibody demonstrates significant advantages in blocking the interaction between TSLP and downstream proteins. The remarkable advantage of this antibody lies in its ability to effectively block the interaction between TSLP and downstream proteins, offering a promising therapeutic approach for TSLP-related diseases, particularly asthma. Further research and optimization efforts will continue to advance the development of targeted antibody therapies against TSLP, providing renewed hope for the treatment of related conditions.

## Materials and methods

### Computer-guided homology modelling and molecular docking

The crystal structure of TSLP (PDB ID: 5J11) [3] was obtained from the Protein Data Bank (https://www.rcsb.org). To generate the 3D theoretical structures of the T6 VH and VL fragments, computer-guided homology modeling was employed using Modeller software. The template used for homology modeling had a sequence identity of 83% (PDB ID: 6B0W). The structures of other mutants (L49Y, H62Y-L49Y95D) were generated using FoldX [28]. Subsequently, molecular dynamics simulations were conducted to obtain stable conformations.

The GROMACS 2019.6 software with the Charmm36 force field was applied to describe the antibody [29,30]. MD simulations were performed in a periodic boundary box using the SPC water model [31]. To neutralize the systems, chloride and sodium ions were randomly added to the simulation box. Energy minimization was carried out using the steepest descent method, followed by equilibration through 100 ps of NVT (Berendsen temperature coupled with constant particle number, volume, and temperature) [32] and 100 ps of NPT (Parrinello–Rahman pressure coupled with constant particle number, pressure, and temperature) [33] simulations at 300 K and 1 bar. The temperature and pressure coupling constants were set at 0.1 and 2.0 ps, respectively. Long-range electrostatic interactions were modeled using the particle mesh ewald algorithm with an interpolation order of 4 and a grid spacing of 1.6 Å [34]. Van der Waals interactions were calculated with a cutoff value of 10 Å. Bond lengths were constrained using the linear constraint solver (LINCS) algorithm [35]. After equilibrating all thermodynamic properties, the molecular system was simulated for 100 ns with a time step of 2 fs, and the coordinates of all models were saved every 100 ps.

The interaction between antibody and antigen was studied using the ZDOCK webserver [36]. The top-ranked output was visualized using PyMol software.

## In silico mutagenesis

To predict the affinity changes between antibodies and antigens upon mutations, two software tools, namely mCSM-PPI2 [13] and Geo-PPI [15] were utilized. The mutation candidates identified from the integrated analysis of these software tools underwent further refinement based on the following screening criteria: (1) Exclusion of mutations to proline, as it is a rigid residue less suitable for the antibody's complementary determining region (CDR); (2) Exclusion of mutations to methionine and tryptophan, as they are prone to oxidation; (3) Exclusion of mutations involving N-linked glycosylation sites (NxT, NxS), asparagine deamidation sites (NS, NG, NH), aspartic acid isomerization sites (DS, DG, DD), and enzyme cleavage sites (DQ, NS); (4) Exclusion of mutations located >3Å away from the antigen surface to account for potential weakened interactions due to excessive distance; (5) Exclusion of mutations with binding energy changes predicted by mCSM-PPI2 less than 1.

## Screening of antibody T6 from fully synthetic human phage antibody libraries

To construct a high-capacity single-chain antibody library, we modified the original phage display vector PHB-1HSCFV. The antibody genes were obtained through PCR amplification and cloned into the PHB-gIII and PHB-pIX display vectors. Multiple rounds of electroporation and Cre-loxP recombination techniques were employed to generate a PHB-pIX-scFv secondary antibody library with a capacity of $1.5 \times 10^{10}$.

The screening process involved coating, blocking, binding, washing, elution, infection, amplification culture, presentation, collection of the secondary library, storage of the secondary library, and determination of the titer of the secondary library. After three rounds of screening, single clones were cultured in 200 μL of 2×YT medium containing 75 μg/mL of C+, 10 μg/mL of T+, and 1% glucose per well in a 96-well plate at 37°C and 220 rpm for 1.5–2 hours, until reaching the logarithmic growth phase. Diluted helper phage (50-fold excess, 80 μL, 6–7 μL of $1 \times 10^{13}$ cfu/mL M13KO7) was then added to 8 mL of 2×YT medium containing 75 μg/mL of C+ and 10 μg/mL of T+ and incubated at room temperature for 15–30 minutes, followed by incubation at 37°C and 150 rpm in 100 μL of 2×YT medium containing 75 μg/mL of C+ and 10 μg/mL of T+ after the addition of 4×K and 4×IPTG (if using IPTG selection) for 1 hour. The induced culture was then incubated overnight at 30°C and 200 rpm.

Single clones were screened using ELISA, and the positive clones were subjected to sequencing, resulting in the identification of the T6 antibody sequence.

## Validation of cellular blockade ability

To investigate the cellular blocking ability of the antibodies, CHO-T/I cells were selected. The TSLPR: IL-7Rα: STAT5 complex was transfected into the cells at a ratio of 1:1:8. After transfection, the cells were subjected to medium replacement and plate coating. Then, the cells were washed twice with serum-free DMEM and incubated in serum-free medium for 8 hours to induce serum starvation. Subsequently, the cells were treated overnight (16 hours) with a combination of TSLP and blocking antibodies to initiate the blocking effect. The supernatant from each well was collected and mixed with 1× PLB lysis buffer, followed by vigorous shaking at 500 RPM on an oscillating shaker for 10 minutes to ensure complete cell lysis. The lysates containing cells were transferred to tubes for further analysis. The detection system consisted of 130 μL, with 100 μL of substrate and 30 μL of sample.

## Supporting information

**S1 Fig. Analysis of the interaction between TSLP and its receptors.** (A) Analysis of the interaction between TSLP and the TSLPR/IL-7Rα complex (PDB ID: 5J11). (B) Analysis of the interaction between AMG157 and TSLP (PDB ID: 5J13).
(TIF)

**S1 Table. Binding sites of TSLPR, IL-7Rα, AMG157, and T6 on TSLP.**
(XLSX)

**S2 Table. Results of the initial alanine scanning of the model by mCSM-PPI2.**
(XLSX)

**S3 Table. Results of the initial alanine scanning of the model using FoldX.**
(XLSX)

**S4 Table. Results of the first round of site-directed mutagenesis predicted by mCSM-PPI2.**
(XLSX)

**S5 Table. The IC50 values of the mutant strains in the first round of mutations.**
(XLSX)

**S6 Table. Energy changes before and after antibody mutations.**
(XLSX)

**S7 Table. Results of the second round of single point mutations predicted by mCSM-PPI2.**
(XLSX)

**S8 Table. The IC50 values of the mutant strains in the second round of mutations.**
(XLSX)

**S9 Table. The IC50 values of the mutant strains in the third round of combinations.**
(XLSX)

## Author Contributions

**Conceptualization:** Zhiwei Sun, Changyuan Yu, Lida Xu.

**Data curation:** Yitong Lv, He Gong, Xuechao Liu.

**Investigation:** Jia Hao, Lei Xu.

**Project administration:** Zhiwei Sun, Lida Xu.

**Writing – original draft:** Yitong Lv.

**Writing – review & editing:** Changyuan Yu, Lida Xu.

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
