## [Decision Letter · Decision Letter 0]

28 Jan 2024

Dear Prof Yu,

Thank you very much for submitting your manuscript "A Dual Computational and Experimental Strategy to Enhance TSLP Antibody Affinity for Improved Asthma Treatment" for consideration at PLOS Computational Biology.

As with all papers reviewed by the journal, your manuscript was reviewed by members of the editorial board and by several independent reviewers. In light of the reviews (below this email), we would like to invite the resubmission of a significantly-revised version that takes into account the reviewers' comments.

We cannot make any decision about publication until we have seen the revised manuscript and your response to the reviewers' comments. Your revised manuscript is also likely to be sent to reviewers for further evaluation.

Sincerely,

Yang Zhang

Guest Editor

PLOS Computational Biology

Nir Ben-Tal

Section Editor

PLOS Computational Biology

Reviewer's Responses to Questions

**Comments to the Authors:**

Reviewer #1: This manuscript reports a dual computational and experimental strategy to enhance TSLP antibody affinity. It uses a variety of tools and techniques to identify new antibodies specific for TSLP and obtain variant antibodies with higher affinity for TSLP through site-directed mutagenesis. Experimental results show that the newly discovered antibody has higher affinity than the benchmark antibody. This research has important implications for promoting the development of targeted therapeutic antibodies.

There are some issues that need to be addressed or clarified.

1. It mentioned that computer-guided homology modeling was used to generate the 3D structure of the T6 antibody variable domain. Which homology modeling method is specifically used? Furthermore, how to ensure/measure the reliability of the predicted structure? This is an important matter because the following process is based on predicted structures. Advanced protein complex structure prediction methods, such as AlphaFold-Multimer and DMFold, provide confidence in the prediction model and may help illustrate the reliability of the prediction model.

2. Both computational tools (mCSM-PPI2 and FoldX) and experimental methods were used to perform alanine scanning. It would be better if the authors could provide performance of alanine scanning using only computational or experimental methods to demonstrate the necessity of combining the two methods. By the way, how did the authors perform the experimental alanine scan using the predicted T6 antibody structure?

3. It mentioned: "Consequently, these mutation points were combined, leading to the discovery that the H62Y-L49Y95D mutated strain exhibited the optimal blocking efficacy." How do you combine these mutation points? H62Y-L49Y95D represents three mutation points. L49Y comes from the first round mutation, and 95D comes from the second round mutation. What about H62Y? It seems that it should also come from the second round of mutation, but each round of mutation only mutates one amino acid at a time, which means that L95D and H62Y belong to two different variants. So how do you combine two mutations in one variant?

4. The authors state: "It was observed that the mutation of leucine to aspartate at position 95 (L95D) in the light chain exhibited a higher blocking efficacy than AMG157, ...". But Figure 5C shows that the AMG157 curve is lower than the L49Y95D curve in most areas. For context, the lower the curve, the greater the blocking efficacy.

5. The IC50 value mentioned on page 10, line 167, seems to be an important measure of affinity and should be briefly described. IC50 values for each variant of T6 and AFM1557 should be provided along with luciferase expression.

6. How to measure the affinity of antibody against TSLP? The authors provide blocking efficacy and IC50 but not explain how they relate to affinity.

7. What is the meaning of OD in Figure 1A? How does it relate to binding epitopes?

8. The S5 table on line 216 on page 13 should be an S6 table.

Reviewer #2: This is a well written manuscript that provides important new insights. However, I was wondering why the authors opted to use transfected CHO to assess the functionality of their antibody and not a cell type that naturally expresses TSLP/R? This would provide much stronger translational value.

**Have the authors made all data and (if applicable) computational code underlying the findings in their manuscript fully available?**

Reviewer #1: None

Reviewer #2: None

PLOS authors have the option to publish the peer review history of their article (what does this mean?). If published, this will include your full peer review and any attached files.

Reviewer #1: No

Reviewer #2: No
---

## [Decision Letter · Decision Letter 1]

10 Mar 2024

Dear Prof Yu,

We are pleased to inform you that your manuscript 'A Dual Computational and Experimental Strategy to Enhance TSLP Antibody Affinity for Improved Asthma Treatment' has been provisionally accepted for publication in PLOS Computational Biology.

Best regards,

Yang Zhang

Guest Editor

PLOS Computational Biology

Nir Ben-Tal

Section Editor

PLOS Computational Biology

Reviewer's Responses to Questions

**Comments to the Authors:**

Reviewer #1: The author responded well to my questions and revised the manuscript.

Reviewer #2: NA

**Have the authors made all data and (if applicable) computational code underlying the findings in their manuscript fully available?**

Reviewer #1: None

Reviewer #2: None

PLOS authors have the option to publish the peer review history of their article (what does this mean?). If published, this will include your full peer review and any attached files.

Reviewer #1: No

Reviewer #2: No

---

## [Editor Report · Acceptance letter]

14 Mar 2024

PCOMPBIOL-D-23-01949R1 

A Dual Computational and Experimental Strategy to Enhance TSLP Antibody Affinity for Improved Asthma Treatment

Dear Dr Yu,

I am pleased to inform you that your manuscript has been formally accepted for publication in PLOS Computational Biology. Your manuscript is now with our production department and you will be notified of the publication date in due course.

With kind regards,

Zsofia Freund
